# The Accuracy of Self-Administered Web- and App-Based Tools for Audiometric Tests—A Systematic Review

**DOI:** 10.3390/audiolres15030073

**Published:** 2025-06-19

**Authors:** Sahuur Abdullahi Ahmed Sheikh Hassan, Uffe Kock Wiil, Ali Ebrahimi

**Affiliations:** SDU Health Informatics and Technology, The Maersk Mc-Kinney Moller Institute, University of Southern Denmark, 5230 Odense, Denmark; sashe20@student.sdu.dk (S.A.A.S.H.); aleb@mmmi.sdu.dk (A.E.)

**Keywords:** hearing loss, audiometry, app-based, web-based, self-administered, tele-audiology, digital audiometry

## Abstract

**Objectives:** This systematic review aimed to evaluate the diagnostic accuracy of self-administered web- and app-based tools for audiometric testing compared to pure-tone audiometry (PTA), the clinical gold standard. **Methods:** Studies were eligible if they involved human participants, evaluated self-administered digital tools for audiometric testing, reported diagnostic accuracy metrics (e.g., sensitivity, specificity, and accuracy), were published between 2014 and 2024, and were written in English. Studies were excluded if they did not compare to PTA, were reviews, or did not assess self-administered tools. MEDLINE, Web of Science, Scopus, and EMBASE were systematically searched throughout November 2024. Study quality was assessed using the QUADAS-2 tool, evaluating four domains: patient selection, index test, reference standard, and flow and timing. Most studies showed some concern for a risk of bias. **Results:** Twelve studies, including a total of 2453 participants and evaluating 15 applications, met the inclusion criteria. The studies reported wide variability in diagnostic accuracy. Sensitivity ranged from 18% to 100%, specificity from 35.5% to 99.1%, and accuracy from 14% to 97.4%. SHOEBOX and Screenout demonstrated high diagnostic accuracy, while other apps showed inconsistent results across studies and settings. Heterogeneity in definitions of hearing loss, test environments, device and headphone types and a lack of standardized reporting limited comparability. Most studies were conducted in non-soundproof environments, and some had unclear or a high risk of bias. **Conclusions:** Self-administered audiometric apps and web tools show promise for remote hearing screening but require further validation and methodological standardization. Clinicians should interpret the results cautiously given the current variability in performance.

## 1. Introduction

Hearing loss is a significant global health issue affecting individuals across all age groups and regions. According to the World Health Organization (WHO), nearly 20% of the global population experiences hearing loss, with an estimated 430 million individuals currently suffering from disabling hearing loss. Projections suggest that this figure will surpass 700 million by 2050 due to demographic changes such as aging populations and increased exposure to noise pollution [1]. Unaddressed hearing impairment can profoundly affect the quality of life, leading to increased risks of social isolation, cognitive decline, depression, and elevated healthcare utilization, including a reported 47% increase in hospitalization rates among affected individuals [2,3].

Pure-tone audiometry (PTA) is widely recognized as the clinical gold standard for assessing hearing function. This traditional method requires highly specialized equipment and trained audiologists operating within soundproof rooms, which are predominantly found in specialized hearing clinic [4]. However, access to PTA can be severely limited due to factors such as geographical isolation, healthcare resource constraints, and prolonged waiting times for specialist referrals. For instance, individuals living in remote or rural areas, or those with limited mobility due to comorbidities or advanced age, often experience significant barriers to receiving timely audiometric evaluations [5,6]. Additionally, during the COVID-19 pandemic, safety measures and restrictions have further limited access to traditional face-to-face audiometric services, underscoring the urgent need for reliable remote healthcare solutions [7].

In response to these accessibility challenges, there has been a substantial increase in the development of digital audiometric tools available via websites (web-based) and mobile applications (app-based). These self-administered tools aim to provide accessible, cost-effective, and user-friendly alternatives to conventional audiometric testing, enabling patients to conduct hearing assessments independently in non-clinical environments such as homes or community settings [8,9]. In Denmark, technological readiness and high digital literacy have made telehealth solutions particularly viable, with recent statistics indicating that 91% of Danes owned computers and 85% used smartphones for personal purposes as of 2020 [10]. Such widespread digital infrastructure supports the feasibility and adoption of tele-audiology services.

Despite the growing popularity and adoption of digital audiometric platforms, significant variations in diagnostic accuracy have been reported across different tools and settings. Previous systematic reviews have highlighted these discrepancies. Bright and Pallawela (2016) revealed that very few audiometric apps available at the time had undergone rigorous validation in peer-reviewed studies, suggesting considerable uncertainty about their reliability [11]. More recently, a meta-analysis by Chen et al. [12] found improved diagnostic accuracy of smartphone-based audiometry, making it particularly useful in settings lacking access to PTA facilities. However, considerable heterogeneity in study methodologies and outcomes was observed, emphasizing the need for standardization and further investigation into factors influencing diagnostic accuracy, such as ambient noise, headphone calibration, and device-specific variations [12].

Given the rapidly evolving digital audiometric landscape and persistent methodological inconsistencies, it remains crucial to systematically assess the current state of evidence regarding the accuracy of self-administered web- and app-based audiometric tools compared to PTA. This systematic review aims to provide an up-to-date and comprehensive synthesis of existing studies, critically appraising diagnostic accuracy metrics and identifying methodological gaps to effectively inform future research and clinical practice.

## 2. Materials and Methods

This review is conducted in accordance with *Joanna Briggs Institute Manuel for Evidence Synthesis*, 2024 Edition [13]. The reporting guideline is followed according to PRISMA (Preferred Reporting Items for Systematic reviews and Meta-Analyses) [14]. The PRISMA 2020 Checklist for this systematic review is available in Appendix A.

### 2.1. Research Question

The PICO mnemonic stands for population, intervention, comparison, and outcome and was used as a guideline to construct a review title and the research question [13]. The aim of this systematic review was to investigate what and how previous studies have examined and evaluated digital solutions for audiometric tests. This was conducted using the following research question:


*How accurate are self-administered web- and app-based tools for audiometric test compared to pure-tone audiometry?*


The PICO elements can be divided into blocks, where the keywords are found using the three-step search strategy described in Section 2.4. The blocks can be seen in Table 1.

### 2.2. Eligibility

The inclusion and exclusion criteria, summarized in Table 2, were selected based on specific considerations to align with the primary objectives of this review.

Studies were included if they focused explicitly on evaluating digital tools for audiometric testing; reported diagnostic accuracy metrics such as sensitivity, specificity, and accuracy; involved human participants; were published in English; and had publication dates between 2014 and 2024 to reflect current technological relevance.

Studies were excluded if they did not directly compare digital tools to pure-tone audiometry (PTA) administered by an audiologist, or if they investigated conditions other than hearing loss, focused exclusively on specific hearing loss types (e.g., high-frequency hearing loss), examined hearing aid fitting applications, evaluated screening programs, or explored the benefits and limitations of smartphones specifically as audiometric testing platforms. Additionally, systematic reviews and studies comparing various audiometric accessories (e.g., headphones) were not considered. Studies that did not emphasize self-administered audiometric tools were also excluded.

### 2.3. Information Sources

To identify potentially relevant documents, the following databases were searched up to November 2024: MEDLINE, Web of Science, Scopus, and EMBASE. The final search string was exported to EndNote and underwent a duplication removal process. The source of the evidence screening process consisted of a title, an abstract, and finally a full-text examination. The main author, SH, conducted the title and abstract screening and reviewed the final retrieved articles. AE approved the selection of articles sought for retrieval, while UKW oversaw and approved the overall review process step by step, ensuring adherence to the predefined methodology.

### 2.4. Search Strategy and Selection Process

The Johannes Briggs Institute’s three-step search strategy was utilized, the first step being an initial search in the databases PubMed and Web of Science [13]. This initial search was followed by an analysis of keywords contained in the title and abstracts of the retrieved papers. A second search was conducted using all identified keyword and Medical Subject Headings (MeSH) terms on the four included databases. Lastly, the reference list of the papers was searched for additional keywords, and gray literature was not included in this review. The final search string for Scopus can be seen below:


*TS = (“hearing loss” OR “hearing disorder” OR “hearing defect” OR “auditory defect” OR “auditory neuropathy” OR “hearing ability” OR “auditory impairment”) AND TS = (“smartphone” OR “online” OR “mobile” OR “digital health” OR “app-based” OR “remote” OR “m-health” OR “telemedicine” OR “application” OR “internet” OR “touchscreen” OR “web-based”) AND TS = (“pure-tone audiometry” OR “PTA” OR “pure tone”) AND TS = (“sensitivity” OR “specificity” OR “accuracy” OR “mean difference” OR “positive predictive value” OR “negative predictive value”)*


The terms were searched for depending on the database, as the title, abstract, keywords, and MeSH, as well as the filters’ language and publication years were used. The main author screened all the identified records for titles and abstracts against the inclusion and exclusion criteria. The reasons for the excluded records were documented.

### 2.5. Data Collection Process and Data Items

Data from studies included in this review were charted using a table designed to investigate the research question. Data was abstracted based on the author, the study design, the population, the average age of and sex of the participants, the clinical setting where the app-based audiometry were performed, the device, headphones, the name of the technology, the tool used to test for hearing, findings, and the definition of hearing loss. This is presented in Section 3.1. Studies evaluating more than one application were grouped together and listed under one study. Regarding diagnostic accuracy data, it represents an average of both ears over several degrees of hearing loss for most studies. Some studies chose to report on the range of diagnostic accuracy. Sensitivity represents the proportion of individuals whose hearing thresholds exceeded the definition for hearing loss on both the self-administered app and PTA (i.e., true positives). Specificity represents the proportion of individuals whose thresholds were below the definition of hearing on both the app and PTA (i.e., true negatives). Accuracy represents the proportion of all tests in which the app’s classification (above or below the threshold) agreed with PTA.

In cases where certain data items were not explicitly reported in the included studies (e.g., participant age, calibration method, or platform type), the information was either extracted from the Appendix A or inferred when clearly described in other parts of the article. If information could not be determined, it was marked as ‘Not reported’ in the tables.

### 2.6. Study Selection

A total of 591 records were identified from four databases and other sources. After duplicate removal, 250 records were title screened. Based on title and abstract screening, 275 records were excluded, with 27 full-text reports to be retrieved and assessed for eligibility. Of these, 8 were unable to be retrieved as the full text was not freely available, leaving 19 reports to be assessed for eligibility. After a full-text review, 2 records were excluded for not being a self-administered application, 1 record was excluded for not being peer-reviewed, and 4 records were excluded for not including diagnostic accuracy metrics, which left 12 reports to be included in this review. The details of the search results are presented in Figure 1.

A detailed list of the excluded full-text articles with reasons for their exclusion was not formally included in this manuscript but is available upon request.

This review was not registered in a protocol registry. Data extraction was initially conducted by one author and independently verified by another author. Discrepancies were resolved through discussion with a third author.

### 2.7. Study Risk of Bias Assessment

The risk of bias for each included study was assessed using the Quality Assessment of Diagnostic Accuracy Studies-2 (QUADAS-2). QUADAS-2 considers four domains, namely patient selection, index test, reference standard, and flow and timing, to assess the risk of bias. The studies were assessed as having either high, low, or uncertain concern of risk, but no overall quality score was generated. If a study is judged as high or unclear in one or more domains relating to either the risk of bias or the risk of applicability, then it is judged as at risk of bias or at risk of applicability. Likewise, if a study is judged as low in all domains, then it is appropriately judged as having a low risk of bias. Studies are not excluded based on the risk of bias, as it is often preferable to review all relevant evidence [15]. A checklist of the quality assessment questions is provided in Table 3.

## 3. Results

The data-charting form and extraction process was developed and conducted by the main author, who determined what and how to extract the data. Data from studies included in this review were charted using a table designed to investigate the research question. The data from the included studies were collected and presented in tables. For studies evaluating more than one application, they were grouped and listed under one study. Regarding diagnostic accuracy findings, it was either assessed for each degree of hearing loss separately (e.g., mild, moderate, and severe) at specific frequencies (e.g., 1000, 2000, 4000, and 800 Hz) or calculated as an average value across all degrees of hearing loss within each study. In studies reporting diagnostic measures by the degree of hearing loss or at specific frequencies, a range of sensitivity, specificity, and accuracy values was presented. The PTAv (pure-tone average) threshold was used to define hearing loss, and in most cases, a result was classified as a true positive when both the self-administered application and the reference pure-tone audiometry result exceeded the predefined threshold for hearing loss. Information that could not be found was filed out with a “-” in the tables.

### 3.1. Study Characteristics

A total of 12 studies met the inclusion criteria, encompassing a combined sample of 2453 participants. The average participant age ranged from 34.23 to 73.64 years. While most studies exclusively targeted adults, four studies [16,17,18,19] also included adolescents. Nine of the twelve studies reported a higher proportion of female participants. However, none of the studies provided detailed data on participants’ educational background or socioeconomic status, representing a limitation in evaluating the broader generalizability of the findings.

To improve interpretability across studies, we referenced international standards for hearing loss classification when summarizing and comparing diagnostic accuracy. For participants aged 8 years and older, we used the World Health Organization (WHO) 2021 guidelines, which define hearing loss categories as normal (0–19 dB HL), mild (20–34 dB HL), moderate (35–49 dB HL), moderately severe (50–64 dB HL), severe (65–79 dB HL), profound (80–94 dB HL), and complete or total (≥95 dB HL) [1]. For participants under 8 years of age, the classification was based on criteria from the American Speech–Language–Hearing Association (ASHA), which provides adjusted thresholds suitable for pediatric populations [20]. When original studies lacked clarity or did not specify a classification system, this was noted and considered a limitation in our synthesis.

The following data from the 12 included studies was reported in Table 4: author, study design, app and operating system, number of participants, tool for testing hearing loss, and diagnostic findings. Furthermore, the clinical setting, population tested, device used in the test, headphones used in the test, and the definition of hearing loss are re-ported in Table 5.

### 3.2. Devices, Platforms, and Testing Conditions

Across the included studies, 15 unique applications were evaluated, 14 app-based and 1 web-based, against PTA as the reference standard. Devices used included smartphones (e.g., iPhone and Samsung), tablets (e.g., iPad and Android), and web interfaces. Notably, five studies (42%) utilized an iPad as the testing device, underscoring its frequent use in digital hearing assessments.

Headphones varied significantly, ranging from commercial earbuds (e.g., Apple and Samsung) to calibrated audiometric headphones (e.g., RadioEar DD450 and Sennheiser HD 202 II). Such variation likely contributed to differences in test performance, as acknowledged in several studies. The operating systems used were predominantly iOS and Android, although a few apps supported both.

Test environments also differed: Eight studies (67%) were conducted in non-soundproof settings (e.g., homes and community clinics), aligning with real-world self-testing conditions. In contrast, only three studies (25%) were performed in soundproof clinical environments. Two studies did not clearly report the testing environment.

### 3.3. Diagnostic Accuracy

The studies showed wide variability in diagnostic outcomes:**Sensitivity** ranged from 18% to 100%.**Specificity** ranged from 35.5% to 99.1%.**Accuracy**, reported in only two studies, ranged from 14% to 97.4%.

High-performing tools included SHOEBOX and Screenout, which consistently demonstrated high sensitivity and specificity across different settings. On the contrary, apps like Uhear and Mimi exhibited substantial variability in diagnostic performance across studies [17,21,22], possibly due to differences in test environments, device types, and headphone calibration. Notably, three studies [9,21,23] reported that these tools were less accurate in detecting high-frequency hearing loss, a known limitation in many self-administered applications.

Hearing loss thresholds varied across the studies, with cut-off points ranging from >20 dB HL to >35 dB HL. While several studies applied threshold values to define the presence or absence of hearing loss [16,23,24], none of the included studies explicitly categorized hearing loss severity (e.g., mild, moderate, and severe) using internationally recognized classification systems such as those by the WHO 2021 [1] or the ASHA [20]. This lack of standardization in classification limits comparability across diagnostic performance metrics and complicates pooled interpretation.

### 3.4. Summary of Finding

While the reviewed self-administered audiometric tools show considerable promise for remote hearing screening, their performance is highly context-dependent. The variation in hardware, software, the environment, and definitions of hearing loss contributes to inconsistencies in diagnostic accuracy. These findings underscore the need for methodological standardization, calibration protocols, and expanded demographic reporting (including education and socioeconomic status) to enhance reliability and generalizability.

A total of 12 studies were included, encompassing 2453 participants across various age groups and clinical settings. The mean age of participants ranged from 34.23 to 73.64 years, with most studies focusing on adult populations. Four studies included both adults and adolescents. Overall, the sex distribution skewed slightly toward female participants: nine of the twelve studies had more women than men. While detailed information on social class and education level was largely absent from the included studies, this gap represents a notable limitation in understanding the broader generalizability of the findings.

Across the studies, 15 applications were evaluated, 14 app-based and 1 web-based, compared against the reference standard, PTA. Most studies utilized smartphones or tablets (e.g., iPad and Samsung Pocket Plus), and a wide variety of headphones was reported, including Apple earbuds, Bluetooth models, and calibrated over-ear options like the RadioEar DD450. Five studies (42%) used an iPad as the testing device, highlighting its frequent use in digital hearing assessments.

In terms of test environments, eight studies (67%) conducted evaluations in non-soundproof settings, reflecting real-world use cases. Only three studies (25%) were conducted in soundproof rooms, typically in clinical environments. This variability in testing conditions contributes to observed diagnostic differences. Furthermore, several studies reported the impact of the headphone type and testing environment on diagnostic performance [9,16,22,23,24,25].

Diagnostic performance varied widely across studies: sensitivity ranged from 18% to 100%, specificity from 35.5% to 99.1%, and overall accuracy from 14% to 97.4%. Apps such as SHOEBOX and Screenout showed consistently high accuracy, while tools like Uhear and Mimi revealed variable performance across different study settings and populations. Notably, three studies [9,21,23], reported that self-administered tools demonstrated reduced accuracy in detecting high-frequency hearing loss, which is a known limitation in many digital audiometric applications.

In summary, while self-administered tools demonstrate potential, heterogeneity in devices, headphone types, test environments, and diagnostic definitions limits direct comparability. Additional data—especially on demographics such as education and socioeconomic status—would further contextualize these findings and support the equitable deployment of digital audiometric tools.

### 3.5. Risk of Bias in Studies

The QUADAS-2 tool was utilized to assess the risk of bias for each of the included studies. A summary of the assessment is provided in Table 6. In terms of the overall risk, there were concerns about the risk of bias for all studies except for [16]. Regarding the risk of applicability, three studies were judged as having high applicability concerns [18,19,26].

**Table 4 audiolres-15-00073-t004:** Characteristics of included studies: design, population, demographics, and clinical setting.

Author [Reference No]	Study Design	Population	Avg. Age (Years)	Sex (M/F)	Participants	Clinical Setting
Rahim et al. [16]	Comparative study	Adult + adolescent	NR	Male (44.4%); Female (55.6%)	133	Non-soundproof room
Anuar et al. [17]	Prospective experimental study	Adult + adolescent	33	Male (52%); Female (48%)	140	Non-soundproof room
Louw et al. [18]	-	Adult + adolescent	37.9	Male (26.4%); Female (73.6%)	1236	Non-soundproof room
Bauer et al. [19]	Cross-sectional, observational	Adult + adolescent	NR	Male (39.5%); Female (60.5%)	185	Soundproof room
Davis et al. [21]	Prospective cohort study	Adult	56.61	Male (51.1%); Female (48.9%)	90	Non-soundproof room
Yesantharao et al. [22]	Prospective cross-sectional	Adult	52.9	Male (32%); Female (68%)	50	Non-soundproof room
Lee et al. [23]	-	Adult	54.3	Male (48.6%); Female (51.4%)	70	Non-soundproof room
Adkins et al. [9]	Prospective within and between subject	Adult	NR	Male (44.6%); Female (56.4%)	39	NR
Chokalingam et al. [24]	Diagnostic study	Adult	34.23	Male (56.3%); Female (43.7%)	119	Soundproof room
Banks et al. [26]	-	Adult	73.64	Male (44%); Female (56%)	50	Non-soundproof room
Génin et al. [27]	Prospective comparative	Adult	NR	Male (40%); Female (60%)	110	Soundproof room
Goh and Jeyanthi [28]	Prospective validation study	Adult	47.6	Male (39%); Female (61%)	231	Non-soundproof room

Abbreviations: M, male; F, female; and NR, not reported.

**Table 5 audiolres-15-00073-t005:** Characteristics of studies: device, headphones, tool category, findings, and definition of hearing loss.

Author [Reference No]	Device	Headphones	App/OS	Test Type	Findings	Definition of Hearing Loss
Rahim et al. [16]	-	-	Screenout, Website	Tone	Sensitivity 90.9% Specificity 98.9%	PTAv > 25 dB
Anuar et al. [17]	iPad mini	-	uHear, iOS	Tone	Sensitivity 54% Specificity 99%	PTAv > 25 dB
Louw et al. [18]	Samsung Pocket Plus S5301	Sennheiser HD 202 II	hearScreen, iOS and Android	Tone	Sensitivity 81.7% Specificity 83.1%	PTAv > 35 dB (adult) PTAv > 25 dB (adolescent)
Bauer et al. [19]	iPod	No headphones used	Ouviu, iOS	Tone	Sensitivity 97.1% Specificity 96.6%	-
Davis et al. [21]	iPad	Apple earbuds	Mimi, iOS and Android	Tone	Sensitivity 18.2–80% Specificity 35.5–97.1%	PTAv > 20 dB
Yesantharao et al. [22]	iPhone X and iPhone 11	Apple earbuds	Mimi, iOS and Android uHear, iOS	Tone	Sensitivity 97.1% Specificity 91.2% Sensitivity 91.3% Specificity 78.0%	PTAv > 20 dB
Lee et al. [23]	iPad mini	Apple earbuds	Care 4 Ear, iOS and Android	Tone	Accuracies: 38.5–97.4%	-
Adkins et al. [9]	iPad	BYZ Stereo Over-Ear and RadioEar DD450 (SHOEBOX)	SHOEBOX, purchased directly Ear Trumpet, iOS	Tone	Accuracy 94% Accuracy 36% Accuracy 14%	PTAv > 25 dB
Chokalingam et al. [24]	Android phone	Samsung EHS61	Hearing test, Android	Tone	Sensitivity 76.26% Specificity 89.99%	PTAv > 25 dB
Banks et al. [26]	iPad	No headphones used	Digital speech hearing screener, iOS	Speech	Sensitivity 87.9–100% Specificity 68.2–81.3%	PTAv > 35 dB
Génin et al. [27]	iPhone 11 and Galaxy S10	Apple earbuds and Samsung earbuds	Audiclick, iOS and Android	Tone	Sensitivity 66.7–90.3% Specificity 84.4–99.1%	PTAv > 25 dB
Goh and Jeyanthi [28]	Android tablet	Bluetooth (SONUP)	SoTone test, iOS and Android	Tone	Sensitivity 92–96% Specificity 79–90%	PTAv > 20 dB

Abbreviations: PTA, Pure-Tone Audiometry and OS, Operating System.

**Table 6 audiolres-15-00073-t006:** Quality assessment results based on the QUADAS-2 tool.

Author	Risk of Bias Concerns				Applicability Concerns		
	Patient Selection	Index Test	Reference Standard	Flow and Timing	Patient Selection	Index Test	Reference Standard
Rahim et al. [16]	Low	Low	Low	Low	Low	Low	Low
Anuar et al. [17]	Low	Unclear	Unclear	Low	Low	Low	Low
Louw et al. [18]	Low	Low	High	High	Low	Unclear	High
Bauer et al. [19]	Low	Unclear	Unclear	Unclear	Low	Unclear	Unclear
Davis et al. [21]	Low	Unclear	Low	Low	Low	Low	Low
Yesantharao et al. [22]	Low	Unclear	Low	Low	Low	Low	Low
Lee et al. [23]	Low	Unclear	Unclear	Low	Low	Low	Low
Adkins et al. [9]	High	Unclear	Unclear	Low	Low	Low	Low
Chokalingam et al. [24]	High	Low	Low	Low	Low	Low	Low
Banks et al. [26]	Unclear	Low	Low	Low	Unclear	Low	Low
Génin et al. [27]	High	Low	Low	High	Low	Low	Low
Goh and Jeyanthi [28]	Low	Unclear	Low	Low	Low	Low	Low

## 4. Discussion

### 4.1. Principal Findings

The 12 included studies employed a range of study designs, including cross-sectional, prospective comparative, diagnostic, and validation studies. Most studies focused on adult populations, while four also included adolescents [16,17,18,19]. A consistent feature across the studies was the use of tone-based audiometry, with sensitivity and specificity as the most commonly reported diagnostic measures. Notably, five studies utilized iPads as the testing device, suggesting a preference for standardized and widely accessible platforms.

A substantial number of studies were conducted in non-soundproof environments, reflecting real-world settings where self-administered tools are likely to be used. While this enhances ecological validity, it introduces challenges regarding ambient noise and its effect on test accuracy. Conversely, studies performed in soundproof rooms may offer more controlled comparisons to PTA but fail to assess home-use feasibility. Multiple studies acknowledged that the testing environment and headphone type significantly impacted diagnostic performance [9,16,22,23,24,26].

Ambient noise, headphone calibration, and device variation were consistently noted as sources of measurement error. While some studies used calibrated over-ear headphones to minimize variability, others relied on commercial-grade earbuds. A lack of standardization in headphone use and the unavailability of calibration protocols for many apps remain critical barriers to clinical-grade accuracy. Moreover, although several applications were available across both iOS and Android, few were validated across both systems, limiting generalizability. The reliance on internet connectivity also emerged as a constraint, especially in underserved or remote settings.

The comparison of the same applications across different studies revealed high variability in performance. Uhear and Mimi, for instance, were each evaluated in multiple studies and showed substantial differences in diagnostic accuracy depending on the testing context [17,21,22]. Additionally, three studies [9,21,23] highlighted that most tools demonstrated reduced sensitivity at higher frequencies, an important clinical gap since high-frequency hearing loss is often an early indicator of auditory decline.

Previous systematic reviews, such as those by Bright and Pallawela [11] and Chen et al. [12], have similarly identified variability in app-based audiometric accuracy. While these earlier reviews focused predominantly on smartphone apps, the current review includes both app- and web-based tools, such as Screenout, and offers a more recent synthesis of evidence. Furthermore, while some earlier reviews included app store searches followed by selective literature screening, our review adopted a methodologically rigorous and transparent search strategy, adhering to PRISMA guidelines.

This review reaffirms existing concerns about methodological inconsistencies across studies, especially regarding definitions of hearing loss, measurement procedures, and data reporting standards. Importantly, it also provides updated evidence to support the clinical utility of digital audiometry in specific use cases while calling attention to the urgent need for validation, calibration, and demographic inclusivity in future work.

### 4.2. Limitations of Evidence

The evidence reported in this review is subject to several important limitations that affect the strength and comparability of the findings. First, the majority of the included studies (11 out of 12) were assessed as having a potential risk of bias in at least one QUADAS-2 domain. While this does not preclude their inclusion, it raises concerns about internal validity and underscores the need for cautious interpretation of the results.

A critical methodological limitation is the inconsistency in how hearing loss was defined and operationalized across studies. The threshold for pure-tone average (PTAv) varied between >20 dB and >35 dB, directly impacting the classification of true positives and negatives. Furthermore, some studies reported diagnostic accuracy by the specific frequency or degree of loss (e.g., mild and moderate), while others reported overall averages, limiting cross-study comparability.

One key limitation across the included studies is the absence of standardized definitions for hearing loss severity. Although some studies used decibel-based thresholds (e.g., >25 or >35 dB HL) to define hearing loss presence [16,23,24], none applied internationally recognized criteria for categorizing severity into mild, moderate, severe, or profound. This inconsistency in classification impairs the comparability of diagnostic accuracy across studies and highlights the need for standardization to support clinical applicability and future meta-analytic efforts.

Additionally, key statistical measures such as confidence intervals, standard deviations, or agreement indices (e.g., Cohen’s Kappa and the Youden Index) were inconsistently reported or omitted altogether. This absence of variability measures reduces transparency and makes it difficult to assess the reliability of the reported diagnostic performance.

Another important limitation was the lack of detailed demographic information in most studies. Variables such as participants’ educational level, socioeconomic status, and ethnicity were largely unreported, hindering the ability to evaluate the generalizability and equity of access in real-world settings.

Finally, several potentially relevant studies were excluded due to restricted access or a lack of diagnostic performance metrics, which may have introduced selection bias and limited the comprehensiveness of the evidence base.

### 4.3. Limitation of the Review Process

This review also has methodological constraints that must be acknowledged. Notably, the title/abstract screening, full-text review, data extraction, and risk of bias assessment were conducted primarily by a single reviewer. Although steps were taken to validate decisions through oversight and selective double-checking by co-authors, the risk of reviewer bias and missed eligibility criteria remains a concern.

The QUADAS-2 tool, while rigorous, is best applied through consensus between multiple assessors to ensure consistency and objectivity. In this review, it was used by a single reviewer, which may have introduced subjective variability in the risk assessments.

Moreover, the exclusion of studies that did not report diagnostic accuracy metrics, while necessary to maintain methodological rigor, may have omitted informative research that could provide context or support for broader interpretations. Similarly, gray literature and non-English studies were not included, potentially narrowing this review’s scope.

Nonetheless, this review followed the PRISMA framework, employed a structured search strategy across four major databases, and applied clear eligibility criteria. These practices support the methodological integrity of the findings despite the limitations noted.

### 4.4. Implication for Practice, Policy, and Future Research

The findings from this review have important implications for clinical practice, healthcare policy, and research. While self-administered audiometric tools offer a promising pathway for increasing access to hearing screening, particularly in underserved or remote populations, their diagnostic accuracy remains variable and context-dependent.

Clinicians and policymakers should be cautious in adopting these tools without local validation. Implementation should be paired with clear usage guidelines, calibration protocols, and patient education to minimize errors and enhance reliability.

From a policy perspective, the integration of self-administered audiometric testing into national screening strategies should prioritize tools that have undergone rigorous validation and provide performance metrics stratified by key subgroups. Additionally, digital health equity must be considered, especially for individuals with limited digital literacy or poor internet access.

Future research should prioritize the following:Standardized definitions and thresholds for hearing loss;Detailed reporting of diagnostic metrics (e.g., confidence intervals and stratified accuracy);Cross-platform and cross-environment validation (e.g., iOS vs. Android and home vs. clinic);Inclusion of socio-demographic data for equity assessment;Evaluation of usability, feasibility, and cost-effectiveness in real-world settings.

A coordinated effort toward methodological standardization and transparent reporting will be essential to advance the field and support evidence-based implementation of digital audiometry tools.

## 5. Conclusions

This systematic review identified 12 studies evaluating 15 web- and app-based tools for audiometric testing, which all were included based on a systematic screening process. These studies were retrieved from four databases and form the basis for answering the following research question: How accurate are self-administered web- and app-based tools for audiometric testing compared to pure-tone audiometry? Diagnostic accuracy values varied between studies, with factors such as the testing environment, the device type, and the thresholds used to define hearing loss differing. The lack of standardized diagnostic criteria, reporting practices, and consistent methodology makes direct comparisons between tools difficult. While several tools show promise for detecting hearing loss in non-clinical settings, further research using standardized diagnostic criteria is needed when evaluating tools.

## Figures and Tables

**Figure 1 audiolres-15-00073-f001:**
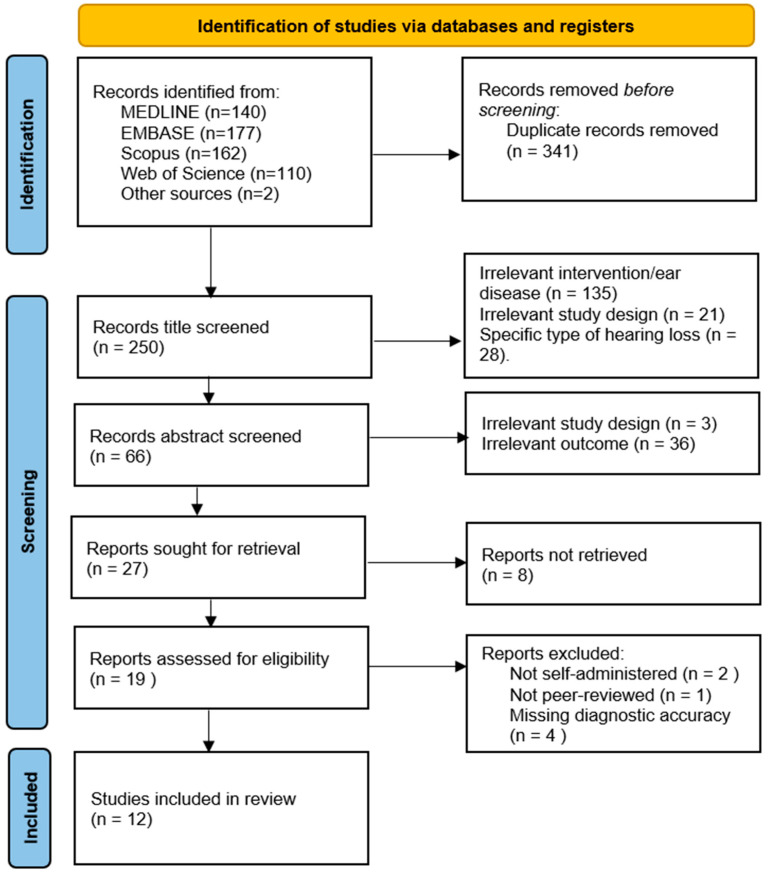
PRISMA 2020 flow diagram. n = number of records.

**Table 1 audiolres-15-00073-t001:** The PICO framework.

Population (P)	Intervention (I)	Comparator (C)	Outcome (O)
Hearing loss	Smartphone	Pure-tone audiometry	Sensitivity
Hearing disease	Online	PTA	Specificity
Hearing defect	Mobile		Accuracy
Auditory defect	Digital health		Mean difference
Auditory neuropathy	App-based		Positive predictive value
Hearing ability	Remote		Negative predictive value
Hearing loss	m-health		
Auditory impairment	Tele		
	Application		
	Internet		
	Cellphone		
	Digital		
	Touchscreen		
	Web-based		

*Pure-Tone Audiometry (PTA).*

**Table 2 audiolres-15-00073-t002:** List of the inclusion and exclusion criteria.

**Inclusion Criteria**
1. The main goal of the study must be examining digital tools for an audiometric test.
2. The study must include diagnostic accuracy measurements (e.g., sensitivity and specificity).
3. The study must involve human participants.
4. The study must be written in English.
5. The study must be published between 2014 and 2024.
**Exclusion Criteria**
1. Studies that do not compare the tool to pure-tone audiometry (PTA)
2. Review articles.
3. Studies that focus on examining the advantages and disadvantages of smartphones as platforms for audiometric tests.
4. Studies comparing different accessories, e.g., headphones.
5. Studies proposing screening programs.
6. Studies that focus on hearing disorders other than hearing loss.
7. Studies researching hearing aid fitting.
8. Studies not focusing on self-administered tools.
9. Studies limited to a specific hearing loss, e.g., high-frequency hearing loss.

**Table 3 audiolres-15-00073-t003:** Quality assessment criteria.

No.	Quality Question
	**Patient selection**
SCQ1	Was a consecutive or random sample of patients enrolled?
SCQ2	Was a case–control design avoided?
SCQ3	Could the selection of patients have introduced bias?
SCQ4	Are there concerns that the included patients do not match the review question?
	**Index test**
SCQ5	Were the index test results interpreted without knowledge of the results of the reference standard?
SCQ6	If a threshold was used, was it prespecified?
SCQ7	Could the conduct or interpretation of the index test have introduced bias?
SCQ8	Are there concerns that the index test, its conduct, or its interpretation differ from the review question?
	**Reference standard**
SCQ9	Is the reference standard likely to correctly classify the target condition?
SCQ10	Were the reference standard results interpreted without knowledge of the results of the index test?
SCQ11	Could the reference standard, its conduct, or its interpretation have introduced bias?
SCQ12	Are there concerns that the target condition as defined by the reference standard does not match the review question?
	**Flow and Timing**
SCQ13	Was there an appropriate interval between index tests and the reference standard?
SCQ14	Did all patients receive a reference standard? Did all patients receive the same reference standard?
SCQ15	Were all patients included in the analysis?
SCQ16	Could the patient flow have introduced bias?

## Data Availability

The data supporting the findings of this review are available from the corresponding author upon reasonable request.

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
