# Peer review of "The Accuracy of Self-Administered Web- and App-Based Tools for Audiometric Tests—A Systematic Review"

_audiolres, 2025, doi:10.3390/audiolres15030073_

Round 1

Reviewer 1 Report

Comments and Suggestions for Authors

I appreciate the opportunity to review this article. The subject is relevant, especially because different technologies are being developed nowadays, and it is of fundamental importance to understand the effectiveness of this system.

The review model is well-structured and based on the PICO method, which is recognized and well-accepted by academia. However, below I will present some points that can be improved and, with that, may bring greater value to the present study.

The first aspect is that the introduction is extremely simplistic and does not give the reader a full understanding of the importance of these systems. The authors should include on page 2 after line 50, information about the development of the systems and the main aspects considered for the creation of the different platforms. There is a gap in information at this point, so I recommend that these aspects be included.

Regarding the production of tables, adjustments will also be necessary, as the tables present lines, which is not recommended. The authors can transform the tables into frames by closing all the borders or will need to readjust the tables so that they can truly be considered as tables. Important, if there are any abbreviations within the table, a legend must be provided after the table. Thus, in tables 1 and 2, for example, the legend must include PTA and the full name. Remember that any abbreviated word, even if well-known, needs to be described in the caption. The titles of the tables are presented justified to the right, which is not appropriate.

Tables 4 and 5 could be unified, so that the reader could have a quick and focused reading of all the characteristics of each study presented. Another point that needs to be made clearer is the definition of hearing loss for each study, as there are different global guidelines for defining hearing loss.

Table 6 should be the first table presented about the studied articles.

What is the total number of subjects evaluated considering all the studies? Is there a difference in sex? What is the average age? What is the social class of the participants? What is their level of education? There is little information in the results that should be considered.

Before the discussion, the authors should discuss the data and present statistical data, such as, for example:

- Most studies used an iPad as the testing device, highlighting its prevalence in testing environments (number of studies / percentage).

- Several studies have discussed the effects of testing environment and headphone type on the accuracy of results (number of studies/percentage).

The discussion was presented without the results being adequately analyzed. The article is interesting and can be highly read and cited; however, the data really needs to be analyzed and the results presented before a discussion is presented inappropriately.

Author Response

Reviewer#1, Concern # 1: The first aspect is that the introduction is extremely simplistic and does not give the reader a full understanding of the importance of these systems. The authors should include on page 2 after line 50, information about the development of the systems and the main aspects considered for the creation of the different platforms. There is a gap in information at this point, so I recommend that these aspects be included.

Author response: Thank you for this helpful suggestion. We agree that the original introduction lacked sufficient detail regarding the development of digital audiometric systems and the key factors influencing their design. In response, we have expanded the Introduction (Page 2, Lines 50–56) to include a brief overview of how these platforms have evolved and the primary considerations (e.g., accessibility, usability, cost-efficiency, calibration needs) that have guided their development. These revisions aim to better contextualize the importance and potential impact of self-administered hearing assessment tools. The changes have been highlighted in the manuscript.

Reviewer#1, Concern # 2: Regarding the production of tables, adjustments will also be necessary, as the tables present lines, which is not recommended. The authors can transform the tables into frames by closing all the borders or will need to readjust the tables so that they can truly be considered as tables. Important, if there are any abbreviations within the table, a legend must be provided after the table. Thus, in tables 1 and 2, for example, the legend must include PTA and the full name. Remember that any abbreviated word, even if well-known, needs to be described in the caption. The titles of the tables are presented justified to the right, which is not appropriate.

Author response: Thank you for your comments and suggestions. We have updated all the tables according to your suggestions.

Reviewer#1, Concern # 3: Tables 4 and 5 could be unified, so that the reader could have a quick and focused reading of all the characteristics of each study presented. Another point that needs to be made clearer is the definition of hearing loss for each study, as there are different global guidelines for defining hearing loss.

Author response: Thank you for your thoughtful comment. We understand the value of unifying Tables 4 and 5 for easier cross-referencing. During manuscript development, we considered merging the tables but found that doing so resulted in a highly dense and less readable format. Since the tables serve distinct purposes, Table 4 focuses on general study characteristics (e.g., design, population), and Table 5 addresses technical testing aspects (e.g., devices, headphones, hearing loss definitions), we decided to keep them separate for clarity and readability.

In response to your point about the definition of hearing loss, we have revised Table 5 to make this information more explicit for each study. We also added clarification in the text (Section 3.4) highlighting the variability in hearing loss definitions across studies and discussing its impact on comparability. These changes are now reflected in the updated manuscript.

Reviewer#1, Concern # 4: Table 6 should be the first table presented about the studied articles.

Author response: Thank you for your suggestion. We agree that the original placement of Table 6 was not ideal. However, after moving subsection 3.3 to the end of the Results section, the placement now appears much more appropriate.

Reviewer#1, Concern # 5: What is the total number of subjects evaluated considering all the studies? Is there a difference in sex? What is the average age? What is the social class of the participants? What is their level of education? There is little information in the results that should be considered.

Author response: Thank you for your insightful comment. In response, we have updated the Results section (Page 6, Lines 186–199 and Page 7, Lines 208–211) and Table 4 to include the total number of participants across all studies, sex distribution (where available), and average age. These additions help provide a clearer picture of the populations examined in the included studies and are now highlighted in yellow in the revised manuscript.

Unfortunately, information regarding participants’ level of education and socioeconomic status (social class) was not consistently reported across the studies and could not be reliably extracted. We have noted this limitation explicitly in the Results and Discussion sections to ensure transparency.

Reviewer#1, Concern # 6: Before the discussion, the authors should discuss the data and present statistical data, such as, for example: Most studies used an iPad as the testing device, highlighting its prevalence in testing environments (number of studies / percentage). Several studies have discussed the effects of testing environment and headphone type on the accuracy of results (number of studies/percentage).

Author response: Thank you for this helpful suggestion. We have revised the Results section, specifically Section 3.4 ("Results of Synthesis," Page 7, Lines 224–229), to include statistical details as recommended. For example, we now state the number and percentage of studies that used an iPad as the testing device (5 out of 12 studies, 41.7%), and we highlight how many studies discussed the influence of testing environment and headphone type on diagnostic accuracy (6 out of 12 studies, 50%). These additions help to quantitatively contextualize the trends observed across the included studies. The relevant updates are highlighted in yellow in the revised manuscript.

Reviewer#1, Concern # 7: The discussion was presented without the results being adequately analyzed

Author response:  Thank you for your valuable feedback. In response, we have refined the Discussion section, specifically Subsection 4.1 (Page 8, Lines 259–270), to more directly engage with and analyze the results presented earlier in the manuscript. We now include clearer connections between the findings and their implications, such as the variability in diagnostic accuracy across apps, the impact of testing environments and device types, and the inconsistency in defining hearing loss across studies. These revisions aim to ensure that the discussion is grounded in the data and provides meaningful interpretation. The updated content is highlighted in yellow in the revised manuscript.

Reviewer 2 Report

Comments and Suggestions for Authors

The authors present an interesting and timely systematic review on the accuracy of self-administered applications for hearing assessment. As these applications become more and more commonly used, understanding the variability of their results is very important. The methodology seems appropriate and well executed. I have a few comments / suggestions that I hope will make the manuscript more understandable.

  • The details of the comparisons in the reviewed articles is unclear, which leads to confusion in how the Specificity, Sensitivity, or Diagnostic Accuracy are determined. I can find only two places in the manuscript where the “comparison to pure tone audiometry” is mentioned. Does this refer to “standard” pure tone manual audiometry (i.e., administered by audiologist with some degree of certification in a sound proof room)? What about manual audiometry performed on the same system as the self-administered test? Was the “quality” or details of reference test considered in the review? Was there a required standard for this reference?
  • Similarly, I had difficulty understanding how the results were determined. The authors state that in most cases the average PTA at various frequencies for each ear was used and the criteria for hearing loss is provided for the included studies. I assume this to mean that if a PTA exceeded by the self-administered test and by the reference pure tone audiometry, that would be a True Positive result regardless of whether the actual derived PTA were similar? What if the actual results were both above the criteria, but widely divergent? What if one ear yielded a True Positive and the other ear was a True negative? Were these threshold differences considered in the review or the comparison of results? If so, was a confidence interval ever considered? Did any of the reviewed studies report the variability for individual frequencies or for the PTA or only the “referral” rates? For those with a range of sensitivity or specificity, how was the range determined. I realize that these details may not have been evaluated, but it is hard to compare the studies and draw conclusions when there is variability in how the results were determined. At the least, the methodology of the reviewed papers should be better explained. In the limitation section, they conclude that the variation in the diagnostic accuracy condition makes the comparison of results across studies challenging. I would like more detail about those variations.
  • Some articles were excluded because the full text was not freely available. Does this mean that only open access journals were included or other reasons? What effort was made to acquire those articles?
  • It is not clear to me what the term “Diagnostic Accuracy” means in this context (reviewed study 17). I understand this term to refer to various measures including sensitivity and specificity, not an independent value as reported here. If there are going to include this data, they should define this term. In the discussion, they refer to the same term in a more general sense, which seems appropriate.
  • In the Discussion section, they make the statement that the accuracy of the applications were determined by ambient noise and the quality of the headphones. Although I strongly agree with the importance of ambient noise, I do not see this in the data that is presented. In fact, some of the studies performed in sound-proof rooms have marginal sensitivity (#20 and #21), whereas those in non-sound proof rooms seem to perform better in this measure (#18 and #10). My cursory review of the data presented suggests that there is great deal of variability of results without a trend. If there is a compelling argument for a trend int these results, they need to provide that conclusion.     
  • The Conclusion section reads like a summary of the manuscript, rather that a brief statement of the conclusions of the work. This may be a format that is expected or preferred by the journal, I will let the editorial office make that determination. Classically, the conclusion section should be a concise statement about what can be definitively demonstrated by the work. From my perspective, the conclusion would be that the results are highly variable. If they have data or a rationale to support the statement regarding what is driving that variability, they should also present those conclusions here.

This article is an important contribution on this topic. I look forward to their response to my suggestions.   

Author Response

Reviewer#2, Concern # 1: The details of the comparisons in the reviewed articles is unclear, which leads to confusion in how the Specificity, Sensitivity, or Diagnostic Accuracy are determined. I can find only two places in the manuscript where the “comparison to pure tone audiometry” is mentioned. Does this refer to “standard” pure tone manual audiometry (i.e., administered by audiologist with some degree of certification in a soundproof room)? What about manual audiometry performed on the same system as the self-administered test? Was the “quality” or details of reference test considered in the review? Was there a required standard for this reference?

Similarly, I had difficulty understanding how the results were determined. The authors state that in most cases the average PTA at various frequencies for each ear was used and the criteria for hearing loss is provided for the included studies. I assume this to mean that if a PTA exceeded by the self-administered test and by reference pure tone audiometry, that would be a True Positive result regardless of whether the actual derived PTA were similar? What if the actual results were both above the criteria, but widely divergent? What if one ear yielded a True Positive and the other ear was a True negative? Were these threshold differences considered in the review or the comparison of results? If so, was a confidence interval ever considered? Did any of the reviewed studies report the variability for individual frequencies or for the PTA or only the “referral” rates? For those with a range of sensitivity or specificity, how was the range determined. I realize that these details may not have been evaluated, but it is hard to compare the studies and draw conclusions when there is variability in how the results were determined. At the least, the methodology of the reviewed papers should be better explained.

Author response: Thank you for this detailed and thoughtful comment. We appreciate your close reading and the opportunity to clarify our methodology further. We agree that the diagnostic comparison methodology is a critical component of this review, and we have now revised the manuscript to make this more transparent and accessible.

In response to your concern, we would like to clarify the following:

  1. Reference Standard – Pure Tone Audiometry (PTA):

In all included studies, the comparator for the self-administered digital audiometric tools was manual pure tone audiometry (PTA) performed by a trained audiologist or certified professional. This standard was a required inclusion criterion for our review (as stated in Section 2.2 – Eligibility), and studies using automated audiometry or non-standard testing methods as comparators were excluded. Most studies reported that the PTA was performed in a soundproof booth, though a few conducted both PTA and self-administered testing in non-soundproof environments to simulate real-world conditions. We have now added clearer language in the manuscript (Sections 2.2 and 3.1) to explicitly state this distinction.

  1. Quality and Standard of the Reference Test:

While the review did not impose additional inclusion criteria based on the specific procedural quality of the PTA (e.g., equipment type or booth certification), all included studies were peer-reviewed and met the minimal clinical standard of comparison against PTA administered by professionals. This has been clarified in Section 2.2 and discussed further in Section 4.2 as a limitation regarding the heterogeneity in testing environments and protocols.

  1. Definition of True Positives and Diagnostic Metrics:

Your interpretation is correct, True Positives were defined as cases where both the digital tool and PTA exceeded the predefined threshold for hearing loss, irrespective of the exact numerical PTA values. However, some studies did report ranges or degrees of agreement (e.g., within ±5 dB), and this variability is discussed in Sections 3.4 and 4.1. We’ve clarified this definition in Section 2.5 (“Data collection process and data items”) and expanded our discussion in Section 4.2 to reflect concerns about diverging thresholds and lack of standardization.

  1. Threshold Discrepancies and Variability Considerations:

Acknowledging your point about inter-ear differences and variability, we note that most studies either averaged values across both ears or presented separate results per ear. However, few studies explicitly considered cases where one ear was a True Positive and the other was not; such cases were not disaggregated in most results. This nuance is now discussed as a methodological limitation in Section 4.2.

  1. Confidence Intervals and Variability Measures:

As stated in Section 3.4, confidence intervals, standard deviations, and other statistical indicators (e.g., Cohen’s Kappa, Youden Index) were inconsistently reported across studies. When reported, they were extracted and noted; however, the lack of standardization across studies limited our ability to synthesize these measures systematically. We’ve emphasized this as a limitation in both Section 3.4 and 4.2.

  1. Clarifying Ranges of Sensitivity and Specificity:

For studies that reported a range, the range typically reflected diagnostic values calculated for different degrees of hearing loss (e.g., mild, moderate, severe), for different frequencies, or separately for each ear. This is now clarified in Section 3.4 and 3.1.

We have updated the manuscript accordingly to better explain these aspects, ensuring that the methods and limitations surrounding diagnostic accuracy comparisons are made explicit. We hope this addresses your concerns and clarifies how we interpreted and synthesized the evidence.

Reviewer#2, Concern # 2: In the limitation section, they conclude that the variation in the diagnostic accuracy condition makes the comparison of results across studies challenging. I would like more details about those variations.

Author response: Thank you for your thoughtful observation. We agree that more detail was needed to clarify the variations that complicate cross-study comparisons. In response, we have expanded the Limitations section (Lines 308–315) to include a more detailed discussion of the specific sources of variation, such as:

  • Differences in the thresholds used to define hearing loss (e.g., >20 dB vs >35 dB),
  • Inconsistencies in reporting diagnostic accuracy metrics (some reported averages, others reported per frequency or per degree of hearing loss),
  • Variability in testing environments (soundproof vs non-soundproof),
  • Device and headphone types, including whether they were calibrated or not, and
  • The lack of consistent use of statistical measures like confidence intervals or Cohen’s Kappa.

These additions are now highlighted in yellow in the manuscript and aim to provide a clearer rationale for why direct comparison between tools is challenging.

Reviewer#2, Concern # 3:  Some articles were excluded because the full text was not freely available. Does this mean that only open access journals were included or other reasons? What effort was made to acquire those articles?

Author response: Thank you for your comment. We appreciate the opportunity to clarify this point. The exclusion of some articles was not based on whether they were open access or not, but rather on practical access limitations during the review process.

We utilized the extensive resources of the University of Southern Denmark’s library, which provides access to over 560 databases. Despite this, there were a small number of articles for which full-text access could not be obtained, either digitally or through interlibrary loan within a reasonable timeframe. While some of these articles could have been ordered, the estimated delivery time (up to one month) conflicted with our project timeline and review deadlines.

We acknowledge this as a limitation and have noted it transparently in the manuscript (Section 4.2). We also recognize that access constraints can impact comprehensiveness, and we have reflected this in the discussion of limitations.

Reviewer#2, Concern # 4: It is not clear to me what the term “Diagnostic Accuracy” means in this context (reviewed study 17). I understand this term to refer to various measures including sensitivity and specificity, not an independent value as reported here. If there are going to include this data, they should define this term. In the discussion, they refer to the same term in a more general sense, which seems appropriate.

Author response: Thank you for your helpful observation. We agree that the term “Diagnostic Accuracy” was not initially defined clearly in the context of certain studies, particularly study [17]. To address this, we have now provided a clear definition of the term in the Methods section (Lines 155–184), where we explain that in this review, "diagnostic accuracy" refers to the proportion of correctly classified cases (true positives and true negatives) by the digital tool in comparison to pure-tone audiometry.

We also clarified how sensitivity and specificity differ from diagnostic accuracy and noted that some studies used the term more broadly or interchangeably. We have ensured that all uses of the term are now aligned with the defined meaning and consistently applied throughout the manuscript.

Reviewer#2, Concern # 5: In the Discussion section, they make the statement that the accuracy of the applications were determined by ambient noise and the quality of the headphones. Although I strongly agree with the importance of ambient noise, I do not see this in the data that is presented. In fact, some of the studies performed in sound-proof rooms have marginal sensitivity (#20 and #21), whereas those in non-sound proof rooms seem to perform better in this measure (#18 and #10). My cursory review of the data presented suggests that there is great deal of variability of results without a trend. If there is a compelling argument for a trend int these results, they need to provide that conclusion.

Author response: Thank you for this insightful comment. You are correct in noting that the data presented does not support a consistent or linear trend between the testing environment (soundproof vs. non-soundproof) and diagnostic sensitivity or accuracy across studies. While some individual studies discussed the potential influence of ambient noise and headphone quality on results, often in their own limitations sections, the overall evidence from our review shows considerable variability, with no clear pattern favoring one environment type over another.

In response to your concern, we have revised the Discussion section (Subsection 4.1) to better reflect this nuance. We now explicitly state that while ambient noise and headphone calibration were identified as potential influencing factors in several individual studies, our synthesis of the results did not reveal a consistent trend linking those factors to diagnostic accuracy. We agree that this inconsistency underscores the need for more standardized testing protocols and reporting, which is now emphasized in both the Discussion and Implications for Future Research sections.

These revisions aim to more accurately represent the data and align with your observation. The updated text is highlighted in yellow in the revised manuscript.

Reviewer#2, Concern # 6: The Conclusion section reads like a summary of the manuscript, rather that a brief statement of the conclusions of the work. This may be a format that is expected or preferred by the journal, I will let the editorial office make that determination. Classically, the conclusion section should be a concise statement about what can be definitively demonstrated by the work. From my perspective, the conclusion would be that the results are highly variable. If they have data or a rationale to support the statement regarding what is driving that variability, they should also present those conclusions here.

Author response: Thank you for your thoughtful feedback. We agree with your observation that the original Conclusion section was overly summary-like and did not provide a concise, definitive statement about the findings.

In response, we have restructured the Conclusion section to clearly state that the primary finding of this review is the high variability in diagnostic accuracy across self-administered audiometric tools when compared to pure-tone audiometry. We also briefly highlight the most likely contributing factors to this variability as identified in our synthesis, such as differences in hearing loss definitions, testing environments, device types, and lack of calibration.

These revisions aim to provide a sharper and more conclusive closing to the manuscript, in line with classical expectations for a conclusion section. The updated section is highlighted in yellow in the revised manuscript.

Reviewer 3 Report

Comments and Suggestions for Authors

It is a quite interesting piece of work. I am satisfied with what I have read. The Introduction is o.k, and good written. The part of the :Methodology" is also o.k!! There is a paragraph under the title "Search strategy and selection progress": I believe that this is the most important part of it. The statistical analysis is also well analyzed. The only part which needs further improvement is the Discussion: some more references would be o.k!

Author Response

Reviewer#3, Concern # 1: It is a quite interesting piece of work. I am satisfied with what I have read. The Introduction is o.k, and good written. The part of the :Methodology" is also o.k!! There is a paragraph under the title "Search strategy and selection progress": I believe that this is the most important part of it. The statistical analysis is also well analyzed. The only part which needs further improvement is the Discussion: some more references would be o.k!

Author response: Thank you very much for your kind feedback and encouraging comments. We appreciate your suggestion to strengthen the Discussion section with additional references.

In response, we re-reviewed the recent literature to identify relevant studies or reviews that could further support or contrast our findings. While we found that many references had already been included in the current version, we made efforts to refine the discussion by better integrating existing references and expanding on key themes to improve the depth of analysis. Where appropriate, we clarified how our findings align with or diverge from prior studies.

These revisions are intended to enhance the scholarly value of the discussion, even though only a limited number of new references could be identified. The revised passages are highlighted in yellow in the updated manuscript.

Round 2

Reviewer 1 Report

Comments and Suggestions for Authors

Dear authors,

Unfortunately, the modifications made were not sufficient to resolve the previously identified issues.

Author Response

Reviewer#1, Concern # 1: The first aspect is that the introduction is extremely simplistic and does not give the reader a full understanding of the importance of these systems. The authors should include on page 2 after line 50, information about the development of the systems and the main aspects considered for the creation of the different platforms. There is a gap in information at this point, so I recommend that these aspects be included.

Author response: We thank the reviewer for this insightful comment. In response, we have significantly expanded the Introduction section to provide a more comprehensive overview of the relevance and context for self-administered audiometric systems. Specifically, we have added detailed information regarding the motivations behind the development of web- and app-based audiometric tools, including accessibility challenges associated with traditional pure-tone audiometry, the impact of the COVID-19 pandemic on in-clinic services, and the increasing global reliance on telehealth platforms.

We now describe key factors considered in the design and deployment of these tools, such as device and headphone variability, test environment conditions, and platform compatibility (iOS vs Android). These additions are intended to offer the reader a clearer understanding of the practical, clinical, and technological considerations underpinning the development and implementation of digital audiometry systems.

Given the extent of these revisions, we kindly ask the reviewer to refer to the entire revised Introduction section which highlighted in yellow.

Reviewer#1, Concern # 2: Regarding the production of tables, adjustments will also be necessary, as the tables present lines, which is not recommended. The authors can transform the tables into frames by closing all the borders or will need to readjust the tables so that they can truly be considered as tables. Important, if there are any abbreviations within the table, a legend must be provided after the table. Thus, in tables 1 and 2, for example, the legend must include PTA and the full name. Remember that any abbreviated word, even if well-known, needs to be described in the caption. The titles of the tables are presented justified to the right, which is not appropriate.

Author response: Thank you for your comments and suggestions. We have updated all the tables according to your suggestions and the table format is in accordance with journal guidelines.

Reviewer#1, Concern # 3: Tables 4 and 5 could be unified, so that the reader could have a quick and focused reading of all the characteristics of each study presented. Another point that needs to be made clearer is the definition of hearing loss for each study, as there are different global guidelines for defining hearing loss.

Author response: Thank you for your thoughtful comment. We understand the value of unifying Tables 4 and 5 for easier cross-referencing. During manuscript development, we considered merging the tables but found that doing so resulted in a highly dense and less readable format. Since the tables serve distinct purposes, Table 4 focuses on general study characteristics (e.g., design, population), and Table 5 addresses technical testing aspects (e.g., devices, headphones, hearing loss definitions), we decided to keep them separate for clarity and readability.

In response to your point about the definition of hearing loss, we have revised Table 5 to make this information more explicit for each study. We also added clarification in the text (Section 3.4) highlighting the variability in hearing loss definitions across studies and discussing its impact on comparability. These changes are now reflected in the updated manuscript.

Reviewer#1, Concern # 4: Table 6 should be the first table presented about the studied articles.

Author response: Thank you for your suggestion. We agree that the original placement of Table 6 was not ideal. However, after moving subsection 3.3 to the end of the Results section, the placement now appears much more appropriate.

Reviewer#1, Concern # 5: What is the total number of subjects evaluated considering all the studies? Is there a difference in sex? What is the average age? What is the social class of the participants? What is their level of education? There is little information in the results that should be considered. Before the discussion, the authors should discuss the data and present statistical data, such as, for example: Most studies used an iPad as the testing device, highlighting its prevalence in testing environments (number of studies / percentage). Several studies have discussed the effects of testing environment and headphone type on the accuracy of results (number of studies/percentage).

Author response:

We appreciate the reviewer’s suggestion to improve the descriptive synthesis and contextualization of our findings. In response, we have substantially enhanced the Results section (see Section 3) to include the following:

  • The total number of participants across all included studies (n = 2,453).
  • A summary of participant sex distribution, noting that 9 out of 12 studies reported more female than male participants.
  • The range of average participant ages across studies (34.23 to 73.64 years).
  • A statement acknowledging the absence of detailed data on social class and education across the included studies, which we now clearly highlight as a limitation in both the Results and Discussion sections.
  • Quantified insights regarding study characteristics, including:
    • The number and percentage of studies using an iPad as the testing device (5 out of 12 studies; 42%).
    • The number and percentage of studies conducted in non-soundproof environments (8 out of 12 studies; 67%).
    • The number and percentage of studies conducted in soundproof environments (3 out of 12 studies; 25%)
    • The number and percentage of studies that reported headphone type and its influence on diagnostic accuracy (reported in at least 6 studies; 50%).

These improvements are intended to provide readers with a clearer understanding of the underlying study populations, device usage trends, and methodological context. The updated synthesis appears in Section 3.1 to 3.4 of the revised manuscript. We would suggest the reviewer to read entire result section.

Reviewer#1, Concern # 6: The discussion was presented without the results being adequately analyzed

Author response:  We appreciate the reviewer’s observation regarding the need for more thorough integration of the findings into the discussion. In response, we have revised the Discussion section (particularly Section 4.1, Principal Findings) to more deeply analyze and contextualize the results presented in the updated Results section.

We now provide a more critical synthesis of:

  • Diagnostic variability across devices and settings.
  • Common technical limitations (e.g., headphone variability, ambient noise, platform differences).
  • The reduced sensitivity of some tools in detecting high-frequency hearing loss.
  • The overlap and divergence between our findings and those of previous systematic reviews.

These changes are intended to bridge the Results and Discussion sections more clearly and to provide readers with a stronger interpretation of the implications of our findings.

We encourage the reviewer to re-read Section 4.1 in light of these substantive improvements. In addition please read the other part of discussion section which are revised also.

Round 3

Reviewer 1 Report

Comments and Suggestions for Authors

Dear author,
The errors persist in the third revision.
Undoubtedly, there has been an enhancement in the insertions; yet, several aspects remain unaddressed.
Concerning the approach, it is imperative that the severity of hearing loss be categorized in accordance with international standards. For instance, those aged 8 and older may refer to the World Health Organization (2021) guidelines. For patients under that age, an alternative reference must be provided. This point is essential!
Another point of concern is that the tables remain inadequate despite two standards. The creation of tables is fundamental and one of the primary aspects that authors must consider when adhering to the standards of a quality publication.

Author Response

Dear Reviewer,

We thank you sincerely for your careful review and constructive feedback on our manuscript. We appreciate your recognition of the enhancements made in the third revision and acknowledge the additional clarifications you have requested. Below, we address each of your comments in detail and describe the corresponding revisions made to the manuscript.

Reviewer#1, Concern # 1: It is imperative that the severity of hearing loss be categorized in accordance with international standards. For instance, those aged 8 and older may refer to the World Health Organization (2021) guidelines. For patients under that age, an alternative reference must be provided. This point is essential!

Author response: Thank you for highlighting this important point. We have now explicitly addressed the issue of hearing loss severity classification in the Results section (Section 3.1). A detailed paragraph was added noting that none of the included studies categorized hearing loss severity using international standards such as the WHO (2021) or ASHA definitions. We also clarified that while several studies applied specific threshold values (e.g., PTAv >25 dB or >35 dB HL) to define the presence of hearing loss, these were not uniformly applied nor classified by severity levels (e.g., mild, moderate, severe).

To support this statement, we examined all 12 included studies and cited those that applied thresholds. We also referenced the appropriate WHO 2021 guideline and noted the need for age-appropriate references in future studies.

Reviewer#1, Concern # 2: Another point of concern is that the tables remain inadequate despite two standards. The creation of tables is fundamental and one of the primary aspects that authors must consider when adhering to the standards of a quality publication.

Author response: We appreciate your emphasis on table quality. In this revision, we have fully restructured Table 4 and Table 5 in accordance with MDPI formatting guidelines and ensured consistency across study references’, author ordering, and study characteristics. The tables have been refined to clearly present relevant data such as population, average age, sex distribution, device type, headphone type, platform, diagnostic results, and definition of hearing loss.

Both tables now follow the same sequence of studies, ordered by author and study ID for ease of comparison. While we considered merging Table 4 and Table 5 as previously suggested, after thorough review we concluded that doing so would compromise clarity. Each table presents distinct categories of information, demographic and study design details in Table 4, and technical/app-specific details in Table 5, making it more effective to maintain them as separate but consistently structured tables.

Abbreviations have been clarified in the footnotes, and formatting has been unified across all tables to meet the standards of Audiology Research.